

# LHC tau-pair production constraints on $a_\tau$ and $d_\tau$

**Ulrich Haisch[1*], Luc Schnell[1,2†] and Joachim Weiss[1,2‡]**

**1** Werner-Heisenberg-Institut, Max-Planck-Institut für Physik,
Föhringer Ring 6, 80805 München, Germany
**2** Technische Universität München, Physik-Department,
James-Franck-Strasse 1, 85748 Garching, Germany

\* haisch@mpp.mpg.de , † schnell@mpp.mpg.de , ‡ jweiss@mpp.mpg.de

## Abstract

We point out that relevant constraints on the anomalous magnetic ($a_\tau$) and electric ($d_\tau$) moment of the tau lepton can be derived from tau-pair production measurements performed at the LHC. Our conclusion is based on the observation that the leading relative deviations from the Standard Model prediction for $pp \to \tau^+\tau^-$ due to $a_\tau$ and $d_\tau$ are enhanced at high energies. Less precise measurements at hadron colliders can therefore offer the same or better sensitivity to new physics with respect to high-precision low-energy measurements performed at lepton machines. We derive bounds on $a_\tau$ and $d_\tau$ using the full LHC Run II data set on tau-pair production and compare our findings with the current best limits on the tau anomalous moments.



## 1 Motivation

Precise measurements of the anomalous magnetic and electric moments of charged leptons, i.e. $a_\ell$ and $d_\ell$, serve as an invaluable tool to test the Standard Model (SM) at the quantum level. They also provide stringent constraints on many scenarios of physics beyond the SM (BSM). For what concerns $a_e$ and $a_\mu$ these tests have reached an impressive relative precision of $3 \cdot 10^{-12}$ [1] and $4 \cdot 10^{-7}$ [2]. Experimental searches for anomalous electric moments of the

electron and muon have not observed any signal, resulting in the upper limit $1.1 \cdot 10^{-29}\, e\,\text{cm}$ [3] and $1.9 \cdot 10^{-19}\, e\,\text{cm}$ [4] on $|d_e|$ and $|d_\mu|$ at 90% confidence level (CL) and 95% CL, respectively.

The relatively short lifetime of the tau lepton makes a direct measurement of its anomalous magnetic or electric moment using the same methods as utilised for the light leptons, such as spin precession methods or spectroscopic methods on trapped particles or bound states, with an accuracy similar to the one obtained in the case of the muon impossible for the foreseeable future. Bounds on $a_\tau$ and $d_\tau$ can therefore only be obtained from processes that involve the production and the decays of tau leptons. In the case of the anomalous magnetic moment for example, the following limits

$$
a_\tau \in
\begin{cases}
[-0.052, 0.013], & \text{DELPHI 95\% CL [5],} \\
[-0.057, 0.024], & \text{ATLAS 95\% CL [6],} \\
0.001^{+0.055}_{-0.089}, & \text{CMS 68\% CL [7],}
\end{cases}
\tag{1}
$$

have been set. The first bound arises from cross-section measurements of photon-induced tau-pair production $\gamma\gamma \to \tau^+\tau^-$ in electron-electron ($ee$) collisions at LEP, while the second and third limits stem from analyses of the same process in lead-lead (PbPb) collisions at the LHC. We add that a global effective field theory analysis of LEP and SLD data [8] leads in comparison to (1) to a tighter limit of $a_\tau \in [-0.007, 0.005]$ at 95% CL. In the case of the tau anomalous electric moment, the current best experimental limits read

$$
\begin{aligned}
\text{Re}(d_\tau) &\in [-1.85, 0.61] \cdot 10^{-17}\, e\,\text{cm}, \\
\text{Im}(d_\tau) &\in [-1.03, 0.23] \cdot 10^{-17}\, e\,\text{cm},
\end{aligned}
\qquad \text{Belle 95\% CL [9].}
\tag{2}
$$

These bounds are based on $e^+e^- \to \tau^+\tau^-$ events collected near the $\Upsilon(4S)$ resonance at the KEKB collider.

In this work we point out that constraints on $a_\tau$ and $d_\tau$ that are competitive or even superior to those quoted in (1) and (2) can be derived from the tau-pair production measurements [10–12] obtained in proton-proton ($pp$) collisions at the LHC Run II. The important observation that leads to this conclusion is that, as a simple consequence of the anomalous moments corresponding to Wilson coefficients of non-renormalisable operators, the contributions of $a_\tau$ and $d_\tau$ to the Drell-Yan (DY) production process $pp \to \tau^+\tau^-$ are enhanced at high energies compared to the SM background. In practice, this energy enhancement turns out to be sufficient to compensate for the lower precision of the $pp$ measurements relative to the $ee$ observables. Our work is organised as follows: in Section 2 we detail the theoretical ingredients that are relevant in the context of this note, while in Section 3 we derive the present constraints on $a_\tau$ and $d_\tau$ that arise from the LHC Run II searches for tau-pair final states. This section also contains a discussion of our results and an outlook.

## 2 Theoretical considerations

The anomalous magnetic and electric moments of the tau lepton can be introduced by considering the gauge-invariant tau-tau-photon vertex up to linear power in the photon four-momentum $q$:

$$
\Gamma_\mu(q^2) = ie\left[ F_1(q^2)\gamma_\mu + \frac{1}{2m_\tau}\left(iF_2(q^2) + F_3(q^2)\gamma_5\right)\sigma_{\mu\nu}q^\nu \right].
\tag{3}
$$

Here $m_\tau \simeq 1.777\,\text{GeV}$ denotes the mass of the tau lepton and $\sigma_{\mu\nu} = i\left(\gamma_\mu\gamma_\nu - \gamma_\nu\gamma_\mu\right)/2$ with $\gamma_\mu$ the usual Dirac matrices. The form factor $F_1(q^2)$ parametrises the vector part of the electromagnetic current and is identified at zero-momentum transfer with the electric charge $e$,

implying $F_1(0) = 1$. The form factors $F_2(q^2)$ and $F_3(q^2)$ are instead related to the anomalous magnetic and electric moment of the tau lepton via

$$a_\tau = F_2(0), \qquad d_\tau = -\frac{e}{2m_\tau} F_3(0).$$ (4)

The SM values of the anomalous magnetic and electric moment of the tau lepton are $a_\tau^{\text{SM}} = 0.0011772$ [13] and $|d_\tau^{\text{SM}}| \simeq 10^{-37}\,e\,\text{cm}$ [14, 15], respectively. Comparing the quoted value of $a_\tau^{\text{SM}}$ to the limits given in (1) one observes that improvements of these bounds by an order of magnitude would make them sensitive to the SM prediction of the tau anomalous magnetic moment. Improvements by twenty orders of magnitude would instead be needed in the case of the tau anomalous electric moment.

The anomalous magnetic and electric moment of the tau lepton can also be parametrised by the SM effective field theory (SMEFT) [16–18] that contains higher-dimensional gauge-invariant operators built from the SM fields. The operators are suppressed by the scale of new physics $\Lambda$ and the leading BSM effects will typically come from the operators of lowest dimension. In the case of $a_\tau$ and $d_\tau$ the relevant effective interactions are of dimension six and encoded in the following Lagrangian:

$$\mathcal{L} = \frac{c_{\tau B}}{\Lambda^2} \left(\bar{L}_L \sigma_{\mu\nu} \tau_R\right) H B^{\mu\nu} + \frac{c_{\tau W}}{\Lambda^2} \left(\bar{L}_L \sigma_{\mu\nu} \sigma^i \tau_R\right) H W^{i,\mu\nu} + \text{h.c.}$$ (5)

Here $L_L = (\nu_{\tau L}, \tau_L)^T$ is the left-handed $SU(2)_L$ third-generation lepton doublet, $\tau_R$ is the right-handed $SU(2)_L$ tau singlet field, $H$ is the Higgs doublet, $B_{\mu\nu}$ and $W_{i,\mu\nu}$ are the $U(1)_Y$ and $SU(2)_L$ field strength tensors and $\sigma^i$ are the Pauli matrices. After electroweak symmetry breaking, the Lagrangian (5) gives rise to the effective interactions

$$\mathcal{L} \supset \frac{c_{\tau\gamma} v}{\sqrt{2}\Lambda^2} \left(\bar{\tau}_L \sigma_{\mu\nu} \tau_R\right) F^{\mu\nu} + \text{h.c.},$$ (6)

where $F_{\mu\nu}$ is the QED field strength tensor and

$$c_{\tau\gamma} = c_w c_{\tau B} - s_w c_{\tau W}.$$ (7)

In terms of the linear combination (7) of Wilson coefficients the tau anomalous magnetic and electric moment can be expressed in the following way:

$$a_\tau = \frac{2\sqrt{2} m_\tau v}{e} \frac{\text{Re}(c_{\tau\gamma})}{\Lambda^2}, \qquad d_\tau = -\sqrt{2} v \frac{\text{Im}(c_{\tau\gamma})}{\Lambda^2}.$$ (8)

Here $c_w \simeq 0.88$ and $s_w \simeq 0.48$ denote the cosine and the sine of the weak mixing angle, respectively, and $v \simeq 246\,\text{GeV}$ is the vacuum expectation value of the Higgs field. Notice that the tau anomalous magnetic (electric) moment is proportional to the real (imaginary) part of the Wilson coefficient (7) showing its CP conserving (violating) character.

In the next section we will derive the present constraints on $a_\tau$ and $d_\tau$ that arise from the LHC Run II searches for tau-pair production. To understand the obtained results qualitatively, we introduce the following ratios of squared Born-level matrix elements

$$\chi_q = \frac{\left|\mathcal{M}_{\text{SM}}(q\bar{q} \to \tau^+\tau^-) + \mathcal{M}_{\text{SMEFT}}(q\bar{q} \to \tau^+\tau^-)\right|^2}{\left|\mathcal{M}_{\text{SM}}(q\bar{q} \to \tau^+\tau^-)\right|^2},$$ (9)

that describe the impact of a non-zero coefficient (7) in $q\bar{q} \to \tau^+\tau^-$ scattering relative to the SM contributions. The relevant tree-level diagrams are shown in Figure 1. Notice that in the SM both $s$-channel photon and $Z$-boson exchange contribute to the scattering,

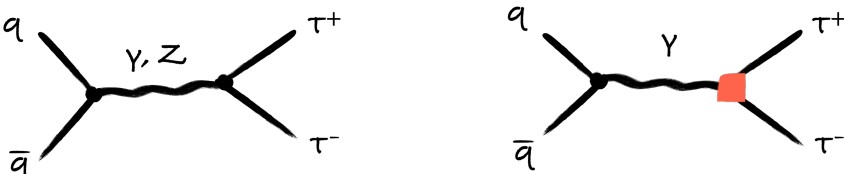

Figure 1: SM (left) and SMEFT (right) contributions to the partonic process $q\bar{q} \to \tau^+\tau^-$. The red box indicates the insertion of the operator with the Wilson coefficient (7). See text for further details.

while in the SMEFT we only consider the photon exchange contribution that is connected to $a_\tau$ and $d_\tau$ via (8) for simplicity. A possible $Z$-boson contribution proportional to the linear combination $c_{\tau Z} = -s_w c_{\tau B} - c_w c_{\tau W}$ of Wilson coefficients is not taken into account since we assume $c_{\tau Z} = 0$. On the technical level, this assumption can be guaranteed by choosing $c_{\tau W} = -t_w c_{\tau B}$ with $t_w \simeq 0.55$ the tangent of the weak mixing angle. Allowing for $c_{\tau Z} \neq 0$ and then deriving constraints in the $c_{\tau\gamma}$–$c_{\tau Z}$ plane from $pp \to \tau^+\tau^-$ would be straightforward, however, we refrain from performing such an analysis in this note. The reason for this is that the bounds (1) and (2) have been derived under the assumption that there is only an anomalous $\gamma\tau^+\tau^-$ coupling but no anomalous $Z\tau^+\tau^-$ coupling. The limits obtained below can therefore be compared directly to (1) and (2) which would not be the case if we were to consider cases with $c_{\tau Z} \neq 0$.

For invariant tau-pair masses sufficiently above the $Z$-pole, i.e. $\hat{s} = m_{\tau\tau}^2 \gg M_Z^2$, we obtain the following approximations for the ratio (9) in the case of down- and up-type initial-state quarks:

$$
\begin{aligned}
\chi_d &\simeq 1 + \frac{64 c_w^4 s_w^4}{e^2 \left(9 c_w^4 - 6 c_w^2 s_w^2 + 25 s_w^4\right)} \left[ \frac{v^2 |c_{\tau\gamma}|^2}{\Lambda^4} \hat{s} - \frac{9e}{8 c_w^2 s_w^2} \frac{v^2 \operatorname{Re}\left(c_{\tau\gamma}\right)}{\Lambda^2} y_\tau \right], \\
\chi_u &\simeq 1 + \frac{256 c_w^4 s_w^4}{e^2 \left(9 c_w^4 + 6 c_w^2 s_w^2 + 85 s_w^4\right)} \left[ \frac{v^2 |c_{\tau\gamma}|^2}{\Lambda^4} \hat{s} - \frac{9e \left(c_w^2 + 5 s_w^2\right)}{16 c_w^2 s_w^2} \frac{v^2 \operatorname{Re}\left(c_{\tau\gamma}\right)}{\Lambda^2} y_\tau \right].
\end{aligned}
\tag{10}
$$

Notice that the first terms in the square brackets of (10), which are due to the interference of the SMEFT contribution with itself, are enhanced by two powers of the tau-pair invariant mass $m_{\tau\tau} = \sqrt{\hat{s}}$. The second terms which arise from the interference of (5) with the SM are instead suppressed by one power of the tau Yukawa coupling $y_\tau = \sqrt{2} m_\tau / v \simeq 7 \cdot 10^{-3}$, which provides the chirality flip needed to obtain a non-zero result. As a result the terms quadratic in $|c_{\tau\gamma}|$ in practice always provide the dominant contribution to $q\bar{q} \to \tau^+\tau^-$ production as far as SMEFT effects are concerned. The results given in (10) hence show that the contributions of $a_\tau$ and $d_\tau$ to the DY production process $pp \to \tau^+\tau^-$ are enhanced at high energies relative to the SM background. Similar observations have been made and exploited for instance also in [19–28].

## 3   Numerical study and discussion

Our calculation of the differential cross-section modifications of tau-pair production due to the anomalous moments of the tau lepton relies on a `FeynRules 2` [29] implementation of the Lagrangian (5) in the `UFO` format [30]. The implementation includes next-to-leading order (NLO) QCD corrections with the relevant counterterms derived by the `NLOCT` package [31]. Our model files are available at [32]. The generation and showering of the sam-

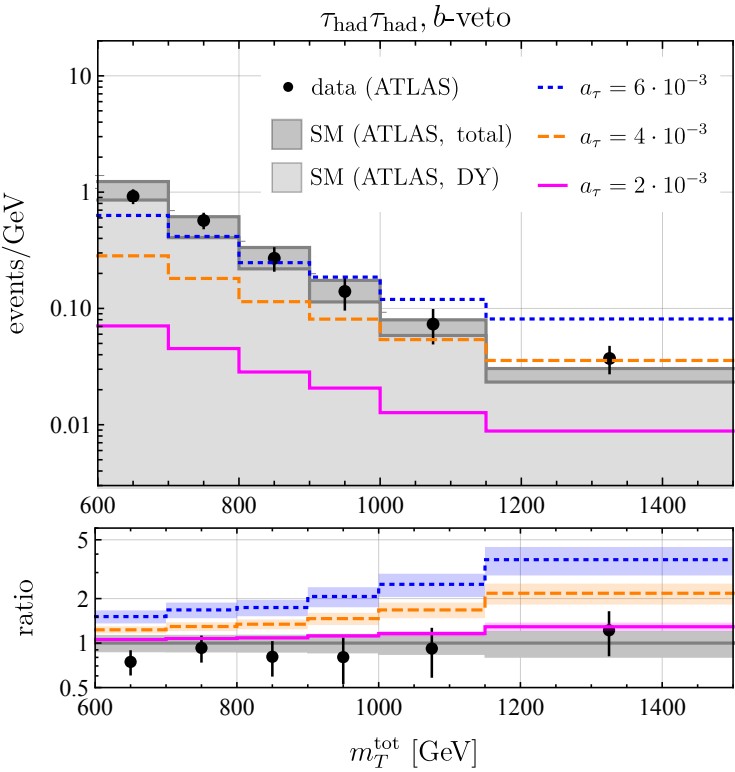

Figure 2: Observed and predicted $m_T^{\text{tot}}$ spectra in the $b$-veto category of the $\tau_{\text{had}}\tau_{\text{had}}$ channel. The black points show the measurements of the ATLAS search [10] with the corresponding statistical uncertainties, the gray (light gray) areas indicate the total expected (DY only) background with the corresponding systematic uncertainties shown in the ratio plot. The dotted blue, dashed orange and solid magenta curves show the expectations of a signal due to $a_\tau = 6 \cdot 10^{-3}$, $a_\tau = 4 \cdot 10^{-3}$ and $a_\tau = 2 \cdot 10^{-3}$, respectively with a systematic uncertainty of 30%. Overflows are included in the last bin of the distributions. For further details consult the main text.

ples is performed with `MadGraph5_aMCNLO` [33] and `PYTHIA 8.2` [34], respectively, using `NNPDF40_nlo_as_01180` parton distribution functions [35]. To improve the statistics in the tails, we use an event generation bias of the form $p_{T,\tau_1}^2/(1\text{TeV})^2$ where $p_{T,\tau_1}$ denotes the transverse momentum of the hardest tau lepton [36].

In order to derive constraints on $a_\tau$ and $d_\tau$ we consider the ATLAS search for hadronic tau ($\tau_{\text{had}}$) leptons [10]. ATLAS identifies the hadronic tau decays by looking for a set of visible decay products in association with missing transverse momentum ($E_{T,\text{miss}}$) from a neutrino. Typically the visible decay products consist of one or three charged pions, called one-track or three-track events, and up to two neutral pions. A seeding jet algorithm [37] is used to reconstructed the $\tau_{\text{had}}$ candidates which are required to have $p_{T,\tau} > 165\,\text{GeV}$ and a pseudorapidity of $|\eta_\tau| < 2.5$ in order to fall into the signal region (SR). One-track (three-track) $\tau_{\text{had}}$ candidates must fulfil "loose" or "medium" tau identification criteria with efficiencies of about 85% (75%) and 75% (60%), respectively. When applied to multijet events the rejection factors of the "loose" or "medium" tau identification are 20 (200) and 30 (500) for one-track (three-track) candidates [37]. In order to end up in the SR, the electric charges of the two $\tau_{\text{had}}$ candidates furthermore have to be opposite and the azimuthal angular difference between them needs to satisfy $|\Delta\phi| > 2.7$. The anti-$k_t$ algorithm with radius $R = 0.4$, as implemented in `FastJet` [38], is used to cluster hadronic jets. These jets are required

to satisfy $p_{T,j} > 20\,\text{GeV}$ and $|\eta_j| < 2.5$. To discriminate between signal and background the ATLAS analysis [10] employs the total transverse mass defined as [39]

$$m_T^{\text{tot}} = \sqrt{m_T^2(\vec{p}_T^{\,\tau_1}, \vec{p}_T^{\,\tau_2}) + m_T^2(\vec{p}_T^{\,\tau_1}, \vec{p}_T^{\,\text{miss}}) + m_T^2(\vec{p}_T^{\,\tau_2}, \vec{p}_T^{\,\text{miss}})}, \tag{11}$$

with the transverse mass of two transverse momenta $p_{T,i}$ and $p_{T,j}$ given by

$$m_T(\vec{p}_T^{\,i}, \vec{p}_T^{\,j}) = \sqrt{2 p_{T,i}\, p_{T,j} (1 - \cos\Delta\phi)}. \tag{12}$$

Here $\tau_1$ ($\tau_2$) denotes the first (second) $\tau_{\text{had}}$ candidate and $\vec{p}_T^{\,\tau_1}$, $\vec{p}_T^{\,\tau_2}$ and $\vec{p}_T^{\,\text{miss}}$ are the vectors with magnitude $p_{T,\tau_1}$, $p_{T,\tau_2}$ and $E_{T,\text{miss}}$. The $E_{T,\text{miss}}$ is constructed from the transverse momenta of all the neutrinos in the event. In the ATLAS search [10] two distinct SRs are defined, one where $b$-jets are vetoed and another one that requires a $b$-jet. The used $b$-tagging working point has a tagging efficiency of 70% and rejections of 9, 36 and 300 for $c$-jets, $\tau_{\text{had}}$ and light-flavoured jets, respectively (cf. [40]). The ATLAS analysis detailed above is implemented into `MadAnalysis 5` [41] which uses `Delphes 3` [42] as a fast detector simulator. Applying our analysis framework to the NLO DY prediction obtained with `MadGraph5_aMCNLO`, we are able to reproduce the SM DY background as given in [10] to within 30%. This comparison serves as a non-trivial crosscheck of our $pp \to \tau^+\tau^-$ analysis.

Distributions of $m_T^{\text{tot}}$ in the $b$-veto category in the final state containing two $\tau_{\text{had}}$ candidates are shown in Figure 2. The black points with error bars correspond to the ATLAS measurements [10] and their statistical uncertainties assuming background only. This search is based on $139\,\text{fb}^{-1}$ of integrated luminosity collected in LHC collisions at $\sqrt{s} = 13\,\text{TeV}$. The gray (light gray) histograms show the expected total (DY) background quoted by ATLAS, the corresponding systematic uncertainties are indicated in the ratio plot of Figure 2. The dotted blue, dashed orange and solid magenta curves instead represent the BSM predictions assuming $a_\tau = 6 \cdot 10^{-3}$, $a_\tau = 4 \cdot 10^{-3}$ and $a_\tau = 2 \cdot 10^{-3}$, respectively, and a systematic uncertainty of 30%. From the figure it is evident that the contributions due to $a_\tau$ are indeed enhanced at high energies with respect to the SM background as argued in Section 2. This enhancement shows up in the tail of the $m_T^{\text{tot}}$ distribution and is rather pronounced even for tau anomalous magnetic moment of $a_\tau = \mathcal{O}(10^{-3})$. For instance, in the bin $m_T^{\text{tot}} \in [1000, 1150]\,\text{GeV}$ the benchmark values of $a_\tau$ indicated in Figure 2 lead to relative enhancements of about 150%, 70% and 16% relative to the SM.

Based on the $\tau_{\text{had}}\tau_{\text{had}}$ search strategy detailed above, we now derive NLO accurate 95% CL limits on $a_\tau$ and $d_\tau$. The significance is calculated as a ratio of Poisson likelihoods modified to incorporate the systematic uncertainties on the background quoted by [10] as well as a 30% systematic uncertainty on our BSM predictions as Gaussian constraints [43]. Our statistical analysis includes the six highest $m_T^{\text{tot}}$ bins of the $b$-veto category, while we ignore the $b$-jet category of [10] because it does not add significance. We obtain

$$|a_\tau| < 1.8 \cdot 10^{-3}, \qquad |d_\tau| < 1.0 \cdot 10^{-17}\, e\,\text{cm}. \tag{13}$$

Comparing these values to the bounds given in (1) and (2) we observe that our limit on the tau anomalous magnetic moment improves significantly on existing limits, while in the case of the tau anomalous electric moment the old and our new constraint are similar in strength. The limits (13) can also be translated into a bound on the Wilson coefficient appearing in (6) and defined in (7). One finds

$$\frac{|c_{\tau\gamma}|}{\Lambda^2} < \frac{1}{(1.5\,\text{TeV})^2}. \tag{14}$$

Let us also spend some words on the future sensitivity of $pp \to \tau^+\tau^-$ searches to the tau anomalous moments at the high-luminosity LHC (HL-LHC) which is expected to collect

$3\,\mathrm{ab}^{-1}$ of integrated luminosity. Assuming a $1/\sqrt{\mathcal{L}}$ scaling of the experimental uncertainties with the luminosity $\mathcal{L}$, which is reasonable as they are statistics dominated in the tail, we find that it may be possible to improve the limits (13) and (14) by a factor of around 2.8. This means that HL-LHC searches for tau-pair production should become sensitive to high-scale BSM contributions that are of the same size as the SM $a_\tau^{\mathrm{SM}} = 0.0011772$ at low energies. Notice that the projected HL-LHC sensitivity is still two orders of magnitude weaker than that of hypothetical Belle II asymmetry measurements in $e^+e^- \to \tau^+\tau^-$ [44–46]. While the latter measurements rely on a polarisation upgrade of the SuperKEKB collider, they could probe $|a_\tau| = \mathcal{O}\left(10^{-6}\right)$.

We conclude this note by commenting on the impact of (13) and (14) on explicit models of BSM physics. The first important remark is that the constraints on $a_\tau$, $d_\tau$ and $c_{\tau\gamma}$ derived in this note only apply to BSM models with new heavy degrees of freedom. In particular, this means that one cannot probe the SM corrections to $a_\tau$ because this contribution will not lead to a quadratic enhancement in the tail of the $m_T^{\mathrm{tot}}$ distribution of the $pp \to \tau^+\tau^-$ process. To understand the generic size of BSM contributions to $a_\tau$, it seems useful to separately discuss the case of models with minimal-flavour violation (MFV) [47] and without. In the case of MFV physics, the possible BSM contributions to the anomalous magnetic moments of the tau and the muon are related via $a_\tau \simeq m_\tau^2/m_\mu^2\, a_\mu \simeq 280\, a_\mu$. Since the measured value of $a_\mu$ cannot deviate from the corresponding SM prediction by much more than $|a_\mu| = \mathcal{O}\left(10^{-9}\right)$ [2,48–50], it follows that MFV deviations in $a_\tau$ cannot exceed $|a_\tau| = \mathcal{O}\left(3\cdot10^{-7}\right)$. The situation is more favourable in BSM models in which the MFV hypothesis is violated, since in such models the top-quark Yukawa coupling can lead to a chiral enhancement of the one-loop contributions [51–54]. For instance, in the case of a scalar $SU(2)_L$ singlet leptoquark with a mass of $2\,\mathrm{TeV}$, it has been shown in [46] that values of $|a_\tau| = 5\cdot10^{-6}$ are possible without violating any direct and indirect constraints. Achieving larger values of $a_\tau$ in non-MFV models might be possible but certainly requires non-trivial model building. In view of this we believe that deviations of $|a_\tau| = \mathcal{O}\left(10^{-5}\right)$ represent a generic upper limit on the possible effects of heavy BSM physics in the anomalous magnetic moment of the tau lepton. Effects of this size easily evade the bounds in (13) and are also too small to be probed using HL-LHC data on tau-pair production. While this is a somewhat chastening conclusion, let us stress again that the search strategy proposed in this note allows to set the best model-independent bound on the effective interactions (6) that exceeds the other existing limits (1) by one order of magnitude.

## Acknowledgments

**Funding information** LS and JW are part of the International Max Planck Research School (IMPRS) on "Elementary Particle Physics". Partial support by the Collaborative Research Center SFB1258 is also acknowledged.

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
