# Peer review of "LHC tau-pair production constraints on $a_\tau$ and $d_\tau$"

_SciPost Physics, doi:SciPost Phys. 16, 048 (2024)_

## Round 1 · Referee Report · Anonymous (Referee 1) · 2023-11-1

Report

This work is precise and succinct, yet detailed enough, highlighting a key insight into probing new physics in anomalous dipole moments of the tau lepton. The authors effectively set limits on the relevant SMEFT dipoles using high-mass Drell-Yan tails. Notably, the determined limits are the most competitive for the anomalous magnetic moment. Before I recommend this for publication, I have two areas I would like the authors to address:

  1. Could the authors briefly comment on the UV models where these bounds are of interest? It is my understanding that perturbative models typically generate these effects at the one-loop level. Referring to eq. (13), this seems pertinent primarily for lighter particle mediators, where the EFT methodology might not be applicable and direct searches come into play.

  2. The authors suggest that the SM target at the HL-LHC will be met. Could they elaborate on the implications for the high-mass tails? Specifically, which SM corrections will become more significant? Additionally, how adequate is the SM prediction at such a precision level?

---

## Round 1 · Referee Report · Anonymous (Referee 2) · 2023-11-2

Report

This article provides a nice analysis of tau pair DY production and its relevance for the tau anomalous couplings. The conclusions rest on a range of assumptions, which are, however, not entirely clear.

(i) all effects are attributed to the tau lepton-sector. How does this compare to the competing coupling modifications that can be expected in other fermion interactions that DY is sensitive to?

(ii) the Z contribution is chosen to vanish. Is this a reasonable assumption? I would expect through Z-photon mixing to see correlated effects away from the Z resonance that can become relevant at large momentum transfers that are highlighted as particularly relevant by the authors.
  • validity: -
  • significance: -
  • originality: -
  • clarity: -
  • formatting: -
  • grammar: -

Author:  Ulrich Haisch  on 2023-11-14  [id 4112]

(in reply to Report 2 on 2023-11-02)
Category:
answer to question

\section*{Report of the referee 2}

\begin{enumerate}

\item[(Q1)] {\it All effects are attributed to the tau lepton-sector. How does this compare to the competing coupling modifications that can be expected in other fermion interactions that DY is sensitive to?}

\item[(A1)] Tau-pair production is a sensitive probe of various operators in the Standard Model effective field theory~(SMEFT). For instance, four-fermion operators of the form $(\bar q \hspace{0.5mm} \Gamma \hspace{0.25mm} q) (\bar \tau \hspace{0.125mm} \Gamma \hspace{0.0mm} \tau)$ with $\Gamma$ denoting a Dirac structure and $q$ a up or down quark are known to lead to visible enhancements in the high-energy tails of the $pp \to \tau^+ \tau^-$ process. See for example

\begin{verbatim}
@article{Greljo:2017vvb,
author = "Greljo, Admir and Marzocca, David",
title = "{High-$p_T$ dilepton tails and flavor physics}",
eprint = "1704.09015",
archivePrefix = "arXiv",
primaryClass = "hep-ph",
reportNumber = "ZU-TH-12-17",
doi = "10.1140/epjc/s10052-017-5119-8",
journal = "Eur. Phys. J. C",
volume = "77",
number = "8",
pages = "548",
year = "2017"
}
\end{verbatim}

for a comprehensive discussion. Notice that the goal of our note is not to perform a global SMEFT fit using the existing ditau data but to point out that the effective interactions~(6) that give a BSM effect to $a_\tau$ and $d_\tau$ also lead to energy-enhanced effects in $pp \to \tau^+ \tau^-$ production. As we show in our work, this opens up the possibility to use existing LHC data on tau-pair production to set bounds on $a_\tau$ that are better than all other existing constraints that are based on the same assumptions.

\item[(Q2)] {\it The $Z$ contribution is chosen to vanish. Is this a reasonable assumption? I would expect through Z-photon mixing to see correlated effects away from the Z resonance that can become relevant at large momentum transfers that are highlighted as particularly relevant by the authors.}

\item[(A2)] We are not exactly sure what the referee implies with the second part of the question. Let us explain our motivation to consider only cases with $c_{\tau Z} =0$ in the analysis. Allowing for $c_{\tau Z} \neq 0$ and deriving constraints in the $c_{\tau \gamma}\hspace{0.25mm}$--$\hspace{0.25mm}c_{\tau Z}$ plane using $pp \to \tau^+ \tau^-$ data would be straightforward, however, we refrain from performing such an analysis in scipost\_202308\_00018v1. The reason for this is simply that the bounds~(1) and~(2) have been derived under the assumption that there is only an anomalous $\gamma \tau^+ \tau^-$ coupling but no anomalous $Z \tau^+ \tau^-$ coupling. The main results of our study,~i.e.~the limits~(13) and (14) that have been derived under the same assumption, can therefore be compared directly to~(1) and~(2) which would not be the case if we were to consider cases with $c_{\tau Z}\neq 0$. We have added a short explanation along these lines to the text.

\end{enumerate}

Attachment:

reply.pdf

---

## Round 2 · List of Changes

Warnings issued while processing user-supplied markup:

  • Inconsistency: plain/Markdown and reStructuredText syntaxes are mixed. Markdown will be used.
    Add "#coerce:reST" or "#coerce:plain" as the first line of your text to force reStructuredText or no markup.
    You may also contact the helpdesk if the formatting is incorrect and you are unable to edit your text.

Reply to report on scipost_202308_00018v1

Ulrich Haisch, Luc Schnell and Joachim Weiss, LHC tau-pair production constraints on $a_\tau$ and $d_\tau$, https://arxiv.org/abs/2307.14133

We thank the referees for their careful reading of the manuscript and for their valuable comments. We try to address all the comments and suggestions in this reply, changing the manuscript accordingly.

Report of the referee 1

(Q1) Could the authors briefly comment on the UV models where these bounds are of interest? It is my understanding that perturbative models typically generate these effects at the one-loop level. Referring to eq. (13), this seems pertinent primarily for lighter particle mediators, where the EFT methodology might not be applicable and direct searches come into play.

(A1) At the very end of our note we have added a paragraph that comments on the impact of our bounds (13) and (14) on explicit models of beyond the Standard Model (BSM) physics. The first important remark is that the constraints derived in our work only apply to models with new heavy degrees of freedom. This for instance means that one cannot probe the Standard Model (SM) corrections to $a_\tau$ because these contributions will not lead to a quadratic enhancement in the tail of the total transverse mass distribution of $pp \to \tau^+ \tau^-$. Similar statements also apply to the corrections in models with axion-like particles or other weakly-coupled BSM theories with light degrees of freedom. Concerning BSM scenarios with heavy new particles, it is useful to distinguish the cases with minimal-flavour violation (MFV) and those without. Since in models with MFV the new-physics effects in $a_\mu$ and $a_\tau$ are strongly correlated, it follows that the existing stringent bounds on new physics in $a_\mu$ limit the possible BSM effects in $a_\tau$. Numerically, one finds that in BSM models with MFV the modifications in $a_\tau$ cannot significantly exceed the level of a few $10^{-7}$. The situation is more favourable in theories with a non-MFV flavour structure because in such models one can have one-loop corrections that are chirally enhanced by the top-quark Yukawa coupling. Models where such an enhancement can be at work are scalar leptoquark~(LQ) scenarios. For instance, in the case of a scalar $SU(2)_L$ singlet LQ with a mass of $2 \, {\rm TeV}$ it has been shown in

@article{Crivellin:2021spu, author = "Crivellin, Andreas and Hoferichter, Martin and Roney, J. Michael", title = "{Toward testing the magnetic moment of the tau at one part per million}", eprint = "2111.10378", archivePrefix = "arXiv", primaryClass = "hep-ph", reportNumber = "PSI-PR-21-27, ZU-TH 56/21", doi = "10.1103/PhysRevD.106.093007", journal = "Phys. Rev. D", volume = "106", number = "9", pages = "093007", year = "2022" }

that values of $|a_\tau| = 5 \cdot 10^{-6}$ are possible without violating any direct and indirect constraint. Achieving larger values of $a_\tau$ in non-MFV models might be possible but certainly requires non-trivial model building. In view of this we believe that deviations of $|a_\tau| = {\cal O} \left ( 10^{-5} \right )$ probably represent a generic upper limit on the possible effects of heavy BSM physics in the anomalous magnetic moment of the tau lepton. Effects of this size easily evade the bounds in (13) and are also too small to be probed using HL-LHC data on tau-pair production. While this is a somewhat chastening conclusion, let us stress again that the search strategy proposed in this note allows to set the best model-independent bound on the effective interactions (6) that by far exceeds the other existing limits (1).

(Q2) The authors suggest that the SM target at the HL-LHC will be met. Could they elaborate on the implications for the high-mass tails? Specifically, which SM corrections will become more significant? Additionally, how adequate is the SM prediction at such a precision level?

(A2) As already explained in (A1) as well as now also in the text, the SM corrections to $a_\tau$ will not lead to quadratically enhanced high-mass tails. The stated possible HL-LHC bound of $|a_\tau| < a_\tau^{\rm SM} = 0.0011772$ hence does neither apply to the SM nor to any other BSM theory with light new degrees of freedom. It only applies to BSM theories with new heavy particles. In our LHC analysis we have incorporated the systematic uncertainties on the background quoted in

@article{ATLAS:2020zms, author = "Aad, Georges and others", collaboration = "ATLAS", title = "{Search for heavy Higgs bosons decaying into two tau leptons with the ATLAS detector using $pp$ collisions at $\sqrt{s}=13$ TeV}", eprint = "2002.12223", archivePrefix = "arXiv", primaryClass = "hep-ex", reportNumber = "CERN-EP-2020-014", doi = "10.1103/PhysRevLett.125.051801", journal = "Phys. Rev. Lett.", volume = "125", number = "5", pages = "051801", year = "2020" }

which amount to around $15\%$ in the phase-space region of interest as well as a $30\%$ systematic uncertainty on our BSM predictions. Notice that the central values and systematic uncertainties on the background distribution of $m_T^{\rm tot}$ as provided by ATLAS are obtained by a simultaneous fit to several control regions. This data-driven method eliminates the need for a precision SM prediction for $pp \to \tau^+ \tau^-$ production because this process is essentially ``measured'' by ATLAS through their fit procedure. Assuming a $1/\sqrt{\cal L}$ scaling of the experimental uncertainties with the luminosity ${\cal L}$, which is reasonable as they are statistics dominated in the tail, we then obtain an improvement factor of around $2.8$ when going from LHC Run II to HL-LHC. We believe that our projection provides a good estimate of the sensitivity of the HL-LHC in probing the effective interactions introduced in (6). Notice that with the better statistics of the HL-LHC, one could include $m_T^{\rm tot}$ bins at higher mass, which might make the limits even stronger than just by a factor 2.8.

Report of the referee 2

(Q1) All effects are attributed to the tau lepton-sector. How does this compare to the competing coupling modifications that can be expected in other fermion interactions that DY is sensitive to?

(A1) Tau-pair production is a sensitive probe of various operators in the Standard Model effective field theory (SMEFT). For instance, four-fermion operators of the form $(\bar q \hspace{0.5mm} \Gamma \hspace{0.25mm} q) (\bar \tau \hspace{0.125mm} \Gamma \hspace{0.0mm} \tau)$ with $\Gamma$ denoting a Dirac structure and $q$ a up or down quark are known to lead to visible enhancements in the high-energy tails of the $pp \to \tau^+ \tau^-$ process. See for example

@article{Greljo:2017vvb, author = "Greljo, Admir and Marzocca, David", title = "{High-$p_T$ dilepton tails and flavor physics}", eprint = "1704.09015", archivePrefix = "arXiv", primaryClass = "hep-ph", reportNumber = "ZU-TH-12-17", doi = "10.1140/epjc/s10052-017-5119-8", journal = "Eur. Phys. J. C", volume = "77", number = "8", pages = "548", year = "2017" }

for a comprehensive discussion. Notice that the goal of our note is not to perform a global SMEFT fit using the existing ditau data but to point out that the effective interactions (6) that give a BSM effect to $a_\tau$ and $d_\tau$ also lead to energy-enhanced effects in $pp \to \tau^+ \tau^-$ production. As we show in our work, this opens up the possibility to use existing LHC data on tau-pair production to set bounds on $a_\tau$ that are better than all other existing constraints that are based on the same assumptions.

(Q2) The $Z$ contribution is chosen to vanish. Is this a reasonable assumption? I would expect through Z-photon mixing to see correlated effects away from the Z resonance that can become relevant at large momentum transfers that are highlighted as particularly relevant by the authors.

(A2) We are not exactly sure what the referee implies with the second part of the question. Let us explain our motivation to consider only cases with $c_{\tau Z} =0$ in the analysis. Allowing for $c_{\tau Z} \neq 0$ and deriving constraints in the $c_{\tau \gamma}\hspace{0.25mm}$--$\hspace{0.25mm}c_{\tau Z}$ plane using $pp \to \tau^+ \tau^-$ data would be straightforward, however, we refrain from performing such an analysis in scipost_202308_00018v1. The reason for this is simply that the bounds (1) and (2) have been derived under the assumption that there is only an anomalous $\gamma \tau^+ \tau^-$ coupling but no anomalous $Z \tau^+ \tau^-$ coupling. The main results of our study, i.e. the limits (13) and (14) that have been derived under the same assumption, can therefore be compared directly to (1) and (2) which would not be the case if we were to consider cases with $c_{\tau Z}\neq 0$. We have added a short explanation along these lines to the text.

We again thank the referees for their very useful feedback and hope that with the above explanations and the implemented changes the manuscript can be published in SciPost in its revised form.

Best regards,

Ulrich Haisch, Luc Schnell and Joachim Weiss

---

## Editorial Decision

published